# Blood Glucose Prediction from Nutrition Analytics in Type 1 Diabetes: A Review

**DOI:** 10.3390/nu16142214

**Published:** 2024-07-10

**Authors:** Nicole Lubasinski, Hood Thabit, Paul W. Nutter, Simon Harper

**Affiliations:** 1Department of Computer Science, The University of Manchester, Manchester M13 9PL, UK; p.nutter@manchester.ac.uk (P.W.N.); simon.harper@manchester.ac.uk (S.H.); 2Diabetes, Endocrine and Metabolism Centre, Manchester Royal Infirmary, Manchester University NHS, Manchester M13 9WL, UK; hood.thabit@mft.nhs.uk; 3Division of Diabetes, Endocrinology and Gastroenterology, School of Medical Science, The University of Manchester, Manchester M13 9NT, UK

**Keywords:** type 1 diabetes, T1D, nutrition, blood glucose prediction, data-driven models, physiological models, hybrid models, PRISMA guidelines, personalized medicine

## Abstract

Introduction: Type 1 Diabetes (T1D) affects over 9 million worldwide and necessitates meticulous self-management for blood glucose (BG) control. Utilizing BG prediction technology allows for increased BG control and a reduction in the diabetes burden caused by self-management requirements. This paper reviews BG prediction models in T1D, which include nutritional components. Method: A systematic search, utilizing the PRISMA guidelines, identified articles focusing on BG prediction algorithms for T1D that incorporate nutritional variables. Eligible studies were screened and analyzed for model type, inclusion of additional aspects in the model, prediction horizon, patient population, inputs, and accuracy. Results: The study categorizes 138 blood glucose prediction models into data-driven (54%), physiological (14%), and hybrid (33%) types. Prediction horizons of ≤30 min are used in 36% of models, 31–60 min in 34%, 61–90 min in 11%, 91–120 min in 10%, and >120 min in 9%. Neural networks are the most used data-driven technique (47%), and simple carbohydrate intake is commonly included in models (data-driven: 72%, physiological: 52%, hybrid: 67%). Real or free-living data are predominantly used (83%). Conclusion: The primary goal of blood glucose prediction in T1D is to enable informed decisions and maintain safe BG levels, considering the impact of all nutrients for meal planning and clinical relevance.

## 1. Introduction

Type 1 Diabetes (T1D) affects more than 9 million people worldwide, with its prevalence increasing by 5% annually [1]. This chronic auto-immune disease results in the inability to produce insulin, requiring self-monitoring and self-management to maintain blood glucose (BG) levels within the optimal range [2,3,4]. Effective BG management requires a balance of BG monitoring, precise exogenous insulin administration (both mealtime and long-acting), appropriate diet intake, and physical activity. Crucial to overall BG management is the management of the mealtime glucose response, the inadequate management of which contributes to a reduced time spent within the target blood glucose range of 3.9–10mmol/L (Time in Range; TIR), leading to an elevated HbA1C (a marker of overall glycemic control) [5,6,7].

BG prediction is a fundamental tool for managing BG levels, resulting in an improvement in the quality of life in people living with T1D (PwT1D) [8]. BG prediction proposes therapeutic solutions, which are critical to the artificial pancreas (AP) systems aiming to optimize BG levels and increase TIR [9]. BG prediction supports disease management through meal planning and insulin dosing suggestions based on BG levels and can be enhanced by including meal composition and volume, previous glycemic excursions, and guidance from the patient’s medical team [10].

Driven by the need to improve BG prediction accuracy and reduce patient burden, there has been extensive research in this field [11]. However, the modeling of the mealtime BG response is complex due to the non-linear nature and magnitude of confounding factors [12,13]. This paper reviews the available literature on the three main primary model strategies (physiological, data-driven, and hybrid models) for BG level prediction T1D, which include nutritional aspects in the prediction model. By analyzing commonly used techniques and the limitations within each of the strategies, the paper aims to provide recommendations for a clinically safe and effective blood glucose prediction model that minimizes user burden while enhancing prediction accuracy using nutritional inputs.

## 2. Related Literature

Work into predicting BG levels at pre-set prediction horizons (PH) [14] has evolved from using historical BG values [15], highlighting the crucial role played by other lifestyle factors, such as nutritional intake in BG dynamics [14,16]. To ensure clinical effectiveness, predictive algorithms must mimic the patient’s unique physiology while adjusting for external factors and be adaptable to the unpredictability of daily life [9,17].

Three approaches have been used to develop BG prediction models, namely, physiological, data-driven, and hybrid prediction models [9]. A prediction model with nutrition inputs at its core would offer an alternative to the three traditional model types. In this light, the authors focus on the role of nutritional factors in enhancing various BG prediction models.

### 2.1. Physiological Blood Glucose Prediction Models

Physiological BG prediction models mimic the glucose regulatory dynamics within various systems of the body [16]. To ensure accurate prediction outputs, the model needs to accurately set the physiological parameters based on a thorough understanding of insulin and glucose metabolism.

This approach necessitates prior knowledge of the underlying principles of action for the physiological and biological systems involved, specifically the glucose–insulin system [9,14,18]. This technique incorporates compartmental models to simulate human BG dynamics and includes additional aspects such as subcutaneous insulin absorption profiles, carbohydrate digestion and absorption, insulin action, and glucose kinetics [9,19].

Physiological BG prediction models are limited by the inherent assumption that the physiological processes controlling the glycemic response to food intake or lifestyle events are reproducible between individuals [12]. Factors that disturb the glucose balance, such as meal intake, are required to be announced for physiological models through self-reporting, which is another limiting aspect of this approach. The inter-patient variability in meal announcement can fluctuate by 30% in the interpretation of the composition of identical meals [20], increasing the need for calibration [21]. Further to the discrepancy in reporting, the inter-patient variability in physiological response makes personalization of physiological models difficult to achieve; in a non-diabetic population, it has been shown that 6% of the mealtime glycemic response variance is linked to person-specific factors [22,23].

### 2.2. Data-Driven Blood Glucose Prediction Models

Data-driven models facilitate BG predictions without prior knowledge, which is key to the physiological prediction models. These models can be based solely on Continuous Glucose Monitor (CGM) inputs, with optional additions of self-reported lifestyle information, such as food intake [24,25], which are the focus of this review. A wide range of data-driven techniques are presented in the literature, with varying core models and additional features utilized. Artificial neural networks (ANNs) have been shown as effective in health-aligned tasks [12] due to their large number of interconnected processing nodes, enabling pattern recognition in complex relationships. Feed-forward neural networks (FFNNs) make up 20% of the reports cited, depending on the features utilized [14].

Time series models rely on historical BG values and utilize linear functions to predict future readings [26]. Regression models utilize statistical algorithms to predict future BG levels based on historical BG values, insulin doses, and other factors, such as physical activity and meal composition. Decision trees, through the utilization of numerical and categorical data, are particularly useful for predicting BG levels based on categorical factors such as food intake, physical activity, and other lifestyle factors [14]. Support Vector Regression (SVR) techniques enable the isolation of patterns not discernible through human analysis, enabling BG prediction based on habitual patterns of human life [27]. The capacity of SVR to handle non-linear relationships makes SVR a promising technique in BG prediction and allows for the personalization of the models, which improves the accuracy of BG prediction. Data-driven models rely on non-physiological formulations to predict and are often built and validated on in silico data, which fail to account for the inter- and intra-subject variability in glucose dynamics at the detriment of physiological meaning [9,28,29]

### 2.3. Hybrid Blood Glucose Prediction Models

Hybrid models for BG prediction represent a fusion of physiological and data-driven methods, leveraging the strengths of both approaches to enhance accuracy. In these models, the physiological components of carbohydrate digestion and absorption, as well as insulin action and absorption profiles, are integrated into the pre-processing stage of the data-driven model [28]. The result is a personalized model that utilizes patient inputs, including self-reported information and mathematical equations, to optimize prediction accuracy [30]. Although this integration increases the complexity of the algorithm, it has the potential to provide more accurate predictions. However, it is important to note that hybrid models can be limited by biases in the training data and the clinical or demographic factors ingrained in the physiological model. Hybrid models utilize simple physiological models that require meal and/or insulin absorption profiles and then fit into a data-driven model to predict future BG values. However, it is important to note that hybrid models can be limited by biases in the training data and the clinical or demographic factors ingrained in the physiological model. These models require parameter tuning, which increases in complexity with multiple models that need to work effectively together to ensure accurate prediction.

## 3. Methodology

The present review followed the Preferred Reporting Items for Systematic Reviews and Meta-Analysis (PRISMA) [31] guidelines to establish a robust methodology. The search was restricted to studies that specifically incorporated nutritional variables as a prominent component in the BG prediction models, given the broad spectrum of input variables that have been utilized in existing prediction models.

### 3.1. Data Sources and Search Strategy

The initial identification of relevant literature was achieved by conducting a comprehensive search through digital libraries, including Google Scholar [32] and PubMed [33]. These platforms were selected as they encompass both the clinical and technical aspects of the inquiry. Specifically, Google Scholar was utilized due to its comprehensive coverage of technical conference papers.

The search criteria employed in this review included the following keywords: “Blood glucose prediction” AND Carbohydrate OR Nutrition OR “Nutritional Information” OR “Mixed Meals” AND “Type 1 Diabetes” OR T1DM.

### 3.2. Eligibility Criteria

The eligibility of articles was as follows:Publications written in English;Publications peer-reviewed and published by 1 May 2024;Publications focused on a Type 1 Diabetes population;Focused on blood glucose prediction algorithms without hyper/hypo-glycemia prevention;Prediction algorithms, including carbohydrate content or meal inputs;Conference papers or academic journal articles.

### 3.3. Study Selection

Once the databases had been queried/searched, Mendeley [34] reference manager was used to remove duplicates. The remaining references were evaluated for inclusion, and this was performed by screening for title and abstract, and finally, the full text was evaluated. The selection process is outlined in the PRISMA flow diagram (Figure 1). Using the framework outlined in Oviedo et al. [9], the features included in the summary tables are presented below. In cases where the information is not explicitly stated, the section is left blank.

Author: This includes the authors listed on the publication, the reference number, and the year of publication.Model Type: The core model used for the prediction is outlined; this includes previously published physiological models and data-driven prediction techniques.Additional Aspects: Any additional techniques used to supplement the core model used.Sub-Systems: This feature in the physiological and hybrid model summary tables highlights any absorption profiles included in the model and details the compartment models presented.Prediction Horizon: Highlights the time frame used in the prediction model.Patients: Here, the population included is outlined. The population is classified as “real” (where the model is validated using real-world data either in an in-patient setting or T1DM outpatients and/or their data) or simulated (where simulated patient data were used for validation).Inputs Used: Here, the checkmark indicates the use of GCM or BG data from self-monitoring blood glucose (SMBG) methods, insulin administration, carbohydrate intake, and/or mixed meals (carbohydrate/protein/fat/fiber).Additional Inputs: Any other included factors are presented here, such as physical activity, insulin-to-carbohydrate ratios, and sleep.Accuracy: Where reported, the models’ performance is indicated, and the metrics used are included along with the measurement used.

## 4. Results

As depicted in Figure 1, the identified unique prediction models (*n* = 138) can be broadly categorized into three types: data-driven blood glucose prediction models (*n* = 74), physiological blood glucose prediction models (*n* = 19), and hybrid blood glucose prediction models (*n* = 45).

A range of prediction horizons is proposed in the literature, with prediction horizons ≤30 min being used in 36% of blood glucose prediction models reported. Prediction horizons of 31–60 min were explored less frequently (34%) than prediction horizons 61–90 min (11%) and 91–120 min (10%) and >120 min (9%).

### 4.1. Physiological Blood Glucose Prediction Models

The review retrieved 19 pieces of work based on physiological models that explicitly used nutritional intakes in the model; a summary of these can be found in Appendix A. The physiological models presented comprise five cornerstone models, along with adaptations and novel approaches, as represented in Figure 2a–d.

Physiological BG prediction models can be classified into two types: “minimal models” and “comprehensive models”. “Minimal models” encompass glucose metabolism and insulin action, while “comprehensive models” incorporate a wide range of intricate physiological variables to mimic real-world scenarios. Figure 2c highlights the aspects included in the physiological models presented in this review.

The incorporation of the rate at which glucose enters the bloodstream is approached in different ways. Notably, the absorption profile of ingested food accounts for carbohydrate absorption at different points throughout the digestive system. Given the parameters of this review, all records incorporated nutritional intake in some regard. Simple carbohydrate intake was featured in 52%, while complex meals were considered in 29% of the reviews. However, other physiological factors that affect absorption profiles, such as glycemic index (14%) and gastric emptying rate (23%), which pertains to the pace at which food is expelled from the stomach into the small intestine, are infrequently integrated as independent features in the physiological models presented in the literature. An alternative rate of glucose appearance, stemming from endogenous glucose production or glycogen conversion, is featured in four of the physiological models identified in this study.

It is noteworthy that the prediction horizon was not explicitly stated in all the identified models. Of those models that specified a prediction horizon, short-term prediction horizons were the most frequently tested, with 31–60 min (29%) being the most common. Mid-term prediction horizons of 61–90 min (18%) and 91–120 min (18%) were also reported in some of the models. Frequently used prediction horizons include ≤30 min (6%) and ≥120 min (6%). Finally, prediction horizons of 24 h are used twice (12%), as is 6 h (12%).

### 4.2. Data-Driven Blood Glucose Prediction Models

The search identified 74 unique records featuring data-driven BG prediction models, as shown in Appendix A. The features included in the returned data-driven blood glucose prediction models are highlighted in Figure 3a–c.

While there is no clear preferred technique, neural networks (NNs) were the most used, with 47% of the prediction models using a form of neural network, followed by Support Vector Regression (SVR) used in 29% of the presented data-driven blood glucose prediction models. Other techniques were less frequently used, the distribution of which is shown in Figure 3a. Several neural network architectures are used in blood glucose prediction, with 25% of the neural networks presented being feedforward neural networks, recurrent neural networks (35%), and LSTM (25%) and convolution neural networks (16%). These algorithms can be trained utilizing various techniques, such as backpropagation, stochastic gradient descent, and other optimization algorithms.

In terms of nutritional inputs, simple carbohydrate intake was the most included variable (72%), with complex meals being used in 28% of the data-driven prediction models presented.

Regarding patient profiles, most studies (83%) used real or free-living data from individuals with diabetes, while simulated data were used in 24% of studies, raising potential issues regarding the generalizability of the results to real-world settings.

The prediction horizon used varied between ≤30 min (45%), 31–60 min (36%), 61–90 min (10%), 91–120 min (8%), and ≥121 min (2%).

### 4.3. Hybrid Prediction Models

The review returned 45 pieces of work proposing hybrid blood glucose prediction models outlined in Appendix A. The physiological models and data-driven techniques utilized, along with nutritional inputs and physiological features proposed by the returned hybrid models, are shown in Figure 4a–d.

Filters were employed in 37.5% of the identified hybrid studies, with the Kalman filter being the most used filter, accounting for 15% of the studies. Bayesian and Gaussian distribution filters were also used in 6% of each study.

Regarding the detail of carbohydrates included, simple carbohydrate intake was the most included aspect (69%), followed by complex meals (26%) and glycemic index (5%). Finally, real/free-living data was the most used patient profile (60%), followed by simulated data (40%).

The prediction horizon used varied between <30 min (31%), 30–60 min (33%), 60–120 min (23%), and >120 min (13%).

## 5. Discussion

The complexity and non-linear nature of the human glucose regulatory system pose a challenge for BG prediction. To ensure an accurate BG prediction that is pragmatically valid, clinically observed trends in glucose metabolism should be considered and incorporated into the model framework [35].

Assessing the accuracy of BG prediction models can be limited by various factors, including the methodology employed, the variation in populations (real versus simulated), and the sample size. The Root Mean Square Error (RMSE) is the gold standard for assessing the accuracy of blood glucose prediction algorithms [36]; alternative assessment methods are also used, including the Clarke Error Grid, which quantifies clinical accuracy [37]. The use of multiple assessment methods allows for comparison between different prediction models [38], which would be otherwise unachievable without running the individual datasets through each presented model.

### 5.1. Validity of Different Model Types

The complexity of T1D and the non-linear nature of BG values present a challenge for accurate BG prediction in real-world settings. With at least 42 contributing factors to glucose dynamics, each model type presents a unique perspective on BG prediction. Physiological models, while perceived as time-consuming [9], allow for the absorption profiles of administered insulin and nutritional intake to be factored in, which are the main contributors to BG fluctuations. The Hovorka model incorporates the effects of meals, physical activity, and insulin doses tuned to the individual, enhancing the accuracy and clinical efficiency, which is potentially the reason it is a frequently used model in physiological and hybrid BG predictions with real-world clinical applications. The Bergman “minimal model” is often used in research and exploratory settings due to its simplicity and ease of use while capturing the essential elements of BG dynamics. Data-driven models, on the other hand, do not require clinical knowledge and allow for predictions to be made from historical information captured from BG sensor devices or manually entered by the user. Hybrid models, as the name suggests, use physiological aspects in the pre-processing stage and use the outputs for the data-driven models.

Given the large number of factors contributing to BG regulation, there needs to be a balance between the complexity of the prediction model and its reliability. The model must be complex enough to offer reliable results while being able to be personalized [10]. The different methods proposed in the literature each have challenges and benefits. There are more data-driven models presented due to the vast array of machine learning techniques available. Making a direct comparison between the techniques would have limited meaning due to the different datasets, prediction horizons, and models used [9].

Neural networks, the most frequently used data-driven technique (47% of returned manuscripts), are trained on extensive datasets encompassing information on meal composition, insulin levels, physical activity, and other physiological parameters, as well as corresponding BG levels. The neural network algorithm can then identify patterns and relationships among these variables and utilize this knowledge to predict future BG levels based on new input data. Additionally, neural networks can learn and adapt to new data over time, allowing for continual improvement of predictions as new data become available [39]. The benefits of using neural networks in BG prediction and one of the reasons for their popularity is their ability to model the complex non-linear nature of BG dynamics. RNNs and LSTMs are able to learn and adapt to new data, which is advantageous in the real world where PwT1D is continuously undertaking new experiences, eating different meals, and providing new data to the prediction model.

In the context of BG prediction, SVR prediction models, used in 22% of all the prediction models presented here and 11% of the hybrid models, work by identifying a hyperplane that best separates the input data into different regions based on their corresponding blood glucose levels. The hyperplane is chosen to maximize the distance between the support vectors, which are the data points closest to the hyperplane. This process, called margin maximization, ensures that the model generalizes well to new data. SVR is particularly useful in situations where the relationship between the input variables, such as historical BG values and nutritional intake, and the output variable, predicted BG levels, is non-linear and complex. SVR uses kernel methods to transform input data into a higher-dimensional space, where the relationship between the variables may be more linear. One advantage of SVR is that it is less susceptible to overfitting than other regression techniques because it uses a regularization parameter to control the complexity of the model. Overall, the use of SVR in BG prediction has shown promising results and is likely to become an increasingly important tool in the development of personalized blood glucose prediction models for individuals with diabetes or other metabolic conditions [40,41].

SVRs are often chosen for the simplicity of the model and the accuracy in smaller datasets, making them beneficial in cases where limited historical BG data are available. The risk of overfitting SVR models can be mitigated with the right choice of kernel and regulation parameters.

One limitation of data-driven models is the quality of the input data, as much of BG management in T1D is reliant on self-reported food intake. Studies have shown that the carbohydrate content of a meal is under-reported by as much as 30% and is frequently underreported at 62.7% of the time, suggesting that two out of three meals have underreported carbohydrate content [20].

One of the key advantages of employing the Kalman filter in BG prediction models is its capacity to handle incomplete or missing data. The recursive algorithm of the Kalman filter updates the state estimates of the system based on new data, even if some measurements are incomplete or missing. This feature can prove especially valuable in situations where CGM data are disrupted or missing due to technical or other factors. Several studies have employed the Kalman filter to enhance the precision of data-driven blood glucose prediction models. Utilizing a Kalman filter to estimate the BG levels of PwT1D by utilizing CGM data enhances the precision of the predictions compared to a standard linear regression model.

### 5.2. Prediction Horizons

In 70% of the studies that specified prediction horizons for all three classifications, the horizons were found to be below 60 min. However, a small percentage (9%) of studies allowed for a prediction horizon above 120 min. It is important to consider the time frames of meal absorption, as discussed in the section below, along with the insulin absorption profiles, in evaluating the clinical relevance of blood glucose predictions at both short- and mid-term horizons.

### 5.3. Impact of Nutritional Intake on Prediction Models

Although isolated carbohydrate intake is typically used as the primary input (71% of returned manuscripts across all prediction model types in this review), this approach neglects other essential factors such as glycemic index, protein, lipids, fiber, and total energy intake. Carbohydrate intake and insulin administration are the primary determinants of BG levels, and their timing and dosing are crucial for optimal BG regulation [42]. Relying solely on carbohydrate intake has its limitations in accurately determining mealtime glycemic response. The absorption time frame of food is determined by the rate of glucose absorption, which varies based on the overall meal composition and can impact the rise and fall of blood glucose levels. Several carbohydrate absorption models have been proposed in the literature, some of which were formulated for non-diabetic populations and adapted for PwT1D. A one-compartment model [43], based on the stomach, does not account for the glycemic index, fiber content, or other nutrients besides carbohydrates, while a two-compartment model [44] that includes intestinal absorption considers the timing of absorption. These compartmental models have been expanded to incorporate the breakdown of lipids, proteins, dietary fiber, and starch into monosaccharides, with five compartments: the stomach, intestine, starch breakdown, intestinal glucose absorption, and gastric emptying [7].

The concept of the glycemic index [4] allows for the quantification of physiological responses to different types of carbohydrates in mealtime conditions. Although not directly prescribed for the nutritional management of T1D, the glycemic index can impact the absorption profile of carbohydrates and the post-prandial glucose response. Foods with a lower glycemic index have been shown to have 15% higher bioavailability compared to those with a higher glycemic index, indicating that higher glycemic index meals are absorbed more quickly and have an impact on immediate post-prandial blood glucose levels [45]. High glycemic index meals are fully absorbed within five hours of consumption, highlighting the importance of considering glycemic index in predicting blood glucose levels and estimating remaining carbohydrates for subsequent meals [46,47].

The inclusion of complex meals as input in BG prediction models accounts for the dynamic and complex nature of BG regulation, influenced by multiple factors such as meal timing and composition, insulin levels, physical activity, and other physiological parameters.

Factoring in the need for nutritional information, the burden on users needs to be factored in. Although there is motivation to report nutritional intake when improved BG control is the outcome, nutrition reporting has been shown to cause stress, and continued retention is limited. Perhaps a BG model that factors in the natural detail of nutrition reporting [48], which reflects the entire meal composition (i.e., a one-word meal tag), could combat this burden and provide the necessary context for an accurate BG prediction.

### 5.4. Population Used

The majority of the studies across all the model types (71% of the returned prediction models) used real or free-living data from individuals with diabetes. Simulated data were also used in a substantial proportion of studies (29% of returned manuscripts), which may have implications for the generalizability of the results to real-world settings. Simulated populations, however, cannot fully mimic additional aspects of BG variations, such as mood and sleep patterns, which factor into the inter-patient variability. Consequently, simulated populations tend to offer smaller RMSE than models tested on real-world patients [7]. Many simulation models for T1D are constructed by amalgamating data from diverse clinical trials with varying participant characteristics, leading to inconsistencies in the data synthesis process. This can introduce biases and inaccuracies in the predictive capabilities of the models. These limitations impact the reliability, generalizability, and clinical applicability of BG prediction models modeled on simulated populations [49].

Overall, these results highlight the diverse range of physiological and data-driven models, filters, and carbohydrate inputs used in hybrid blood glucose prediction models, as well as the importance of considering the patient profile of the datasets used. Further research is needed to determine the optimal combinations of these elements for accurate and reliable blood glucose prediction in different patient populations and settings.

The dataset utilized is crucial to ensuring the generalizability of BG prediction models. Small datasets potentially lead to misleading trends and decreased generalizability. Different model types are more prone to over-fitting. However, because of individual physiological differences in BG regulation, a BG prediction model needs to be tailored to the individual. Even more important than the dataset itself is the amount of data or training time used to develop these models. Insufficient data or training can result in the model performing poorly in real-world or unforeseen situations.

## 6. Limitations and Future Work

While systematic reviews offer a powerful tool for synthesizing the available research in the area, there are limitations. There is inherent publication bias, where studies with positive and significant results tend to be published, leading to an unbalanced perspective of the research field. In addition, there is bias in the review process, including the selection of studies, data extraction, and the interpretation of the results. This review focused solely on blood glucose prediction models, which included nutritional inputs, limiting the generalizability of blood glucose predictions.

## 7. Conclusions

The primary goal of blood glucose prediction in the context of T1D is to enable individuals to make informed decisions and maintain a safe BG level, particularly during the period between meals. Given that individuals consume food rather than individual macronutrients, it is crucial to consider the impact of other nutrients on BG levels both in the immediate timeframe and in nutrients on board at the next meal. In real-world scenarios, where individuals consume food every three to four hours, the ability to provide longer prediction horizons can facilitate future meal planning and have clinical relevance in an outpatient setting.

## Figures and Tables

**Figure 1 nutrients-16-02214-f001:**
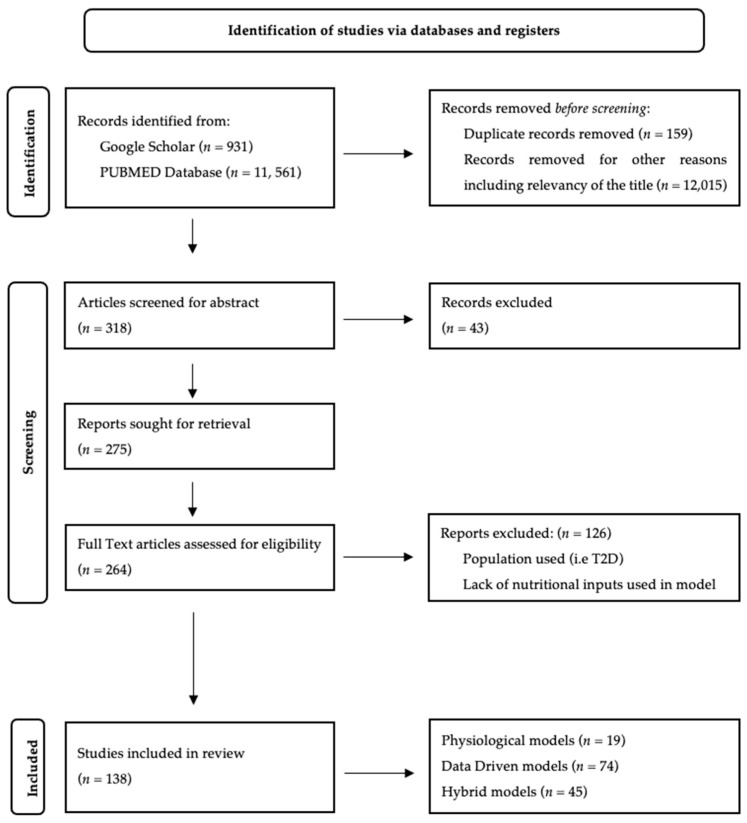
Flow diagram for the identification of selected studies based on the PRISMA guidelines [31].

**Figure 2 nutrients-16-02214-f002:**
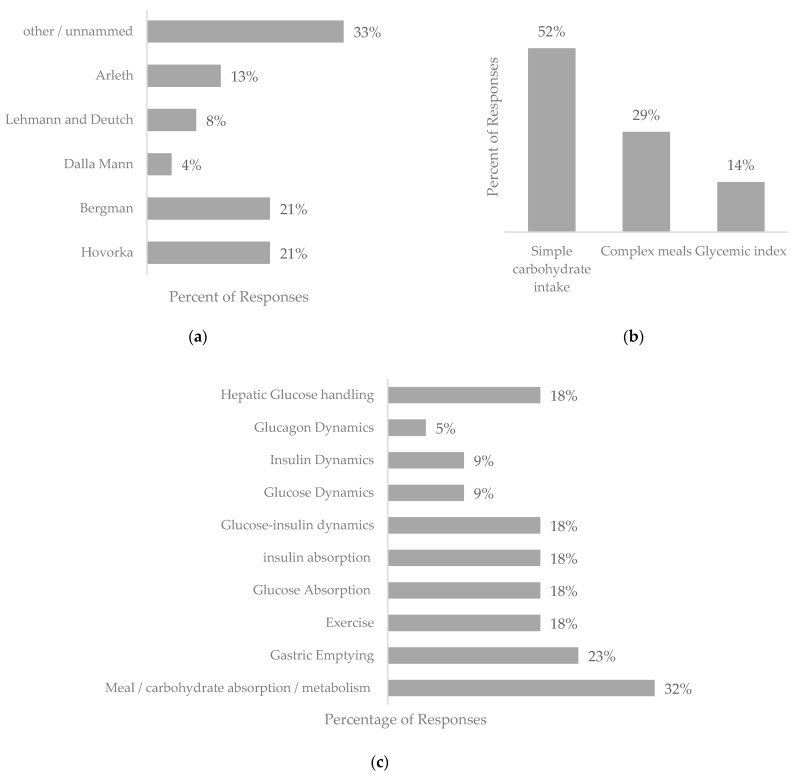
Graphs depicting the features included within the physiological blood glucose prediction models returned in this systematic review. (**a**) Indication of the models represented within the physiological models presented. (**b**) Representation of nutritional input type in physiological models. (**c**) Representation of physiological aspects included in physiological models. (**d**) Representation of population type in physiological models.

**Figure 3 nutrients-16-02214-f003:**
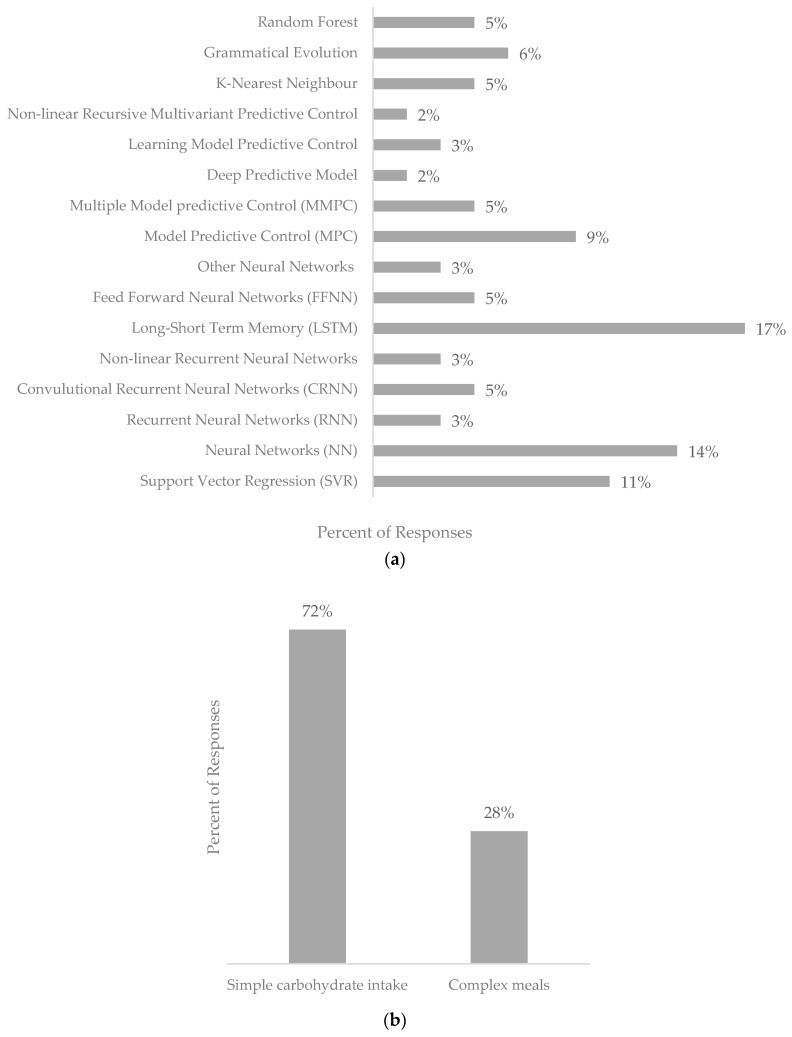
Graphs depicting the features included within the data-driven blood glucose prediction models returned in this systematic review. (**a**) Representation of techniques used in data-driven blood glucose models. (**b**) Representation of nutritional inputs featured in data-driven blood glucose prediction models. (**c**) Representation of populations used in hybrid blood glucose prediction models.

**Figure 4 nutrients-16-02214-f004:**
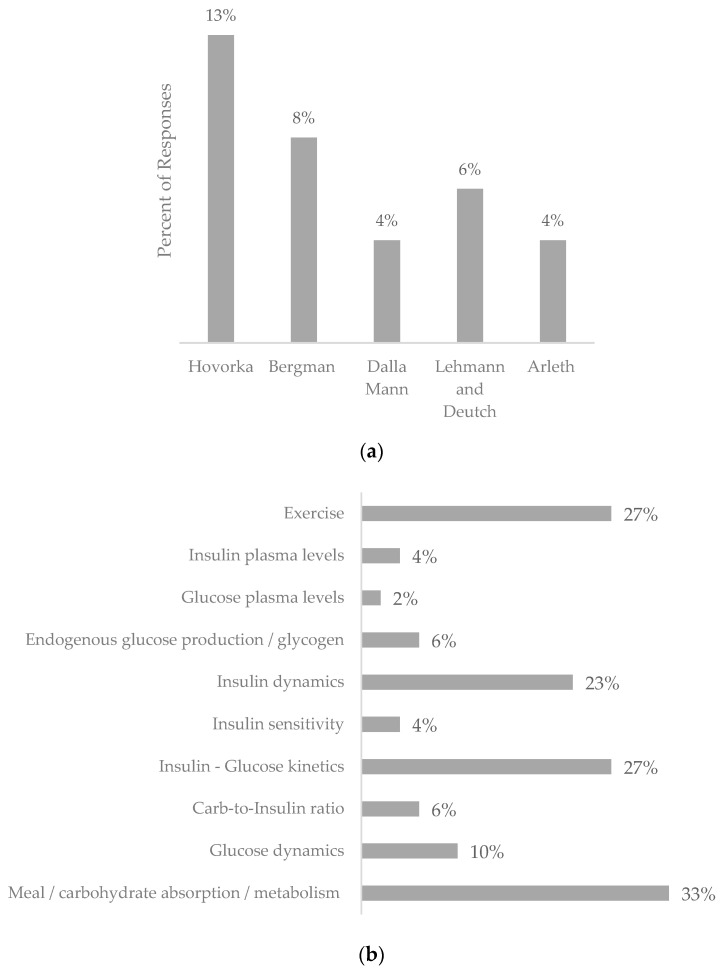
Graphs depicting the features included within the hybrid blood glucose prediction models returned in this systematic review. (**a**). Representation of physiological models used in hybrid blood glucose prediction models. (**b**). Representation of physiological features used in hybrid blood glucose prediction models. (**c**). Representation of the data-driven models used in hybrid blood glucose prediction models. (**d**). Representation of nutritional inputs featured in hybrid blood glucose prediction models.

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
