# Peer review of "Blood Glucose Prediction from Nutrition Analytics in Type 1 Diabetes: A Review"

_nutrients, 2024, doi:10.3390/nu16142214_

Round 1

Reviewer 1 Report

Comments and Suggestions for Authors

See attached. 

Reviewer 2 Report

Comments and Suggestions for Authors

Report on the manuscript “Blood Glucose Prediction from Nutrition Analytics in Type 1 Diabetes: A Review.”

The topic of the manuscript is undoubtedly relevant. The authors use a progressive and reliable method of searching, sifting, and analyzing literature sources. I read the review with great interest and not without benefit. I hope this review will be useful to potential readers. However, I would recommend the manuscript for publication only after major revision.

Main comment

The authors impartially and objectively present scientometric information as a fact in approximately the following style:

Method A was used in 85%, Method B in 10%, and Method C in 5% of publications.

(Sorry, but I simplified everything for clarity. Method can also mean a model, a technique, or something else.)

I ask the authors to expand the analysis. I want to know their opinion on why method A was used in 85% of cases. Is it because method A is easy to implement, or is very accurate, or was the first to be discovered, etc.? Is Method C very promising and has a great future, but its initial rating is very low?

I understand that 85% and 10% are hard facts. I understand that the authors' opinion is subjective. However, I believe that this opinion will help the readers to navigate the ocean of publications.

I am confident that this kind of analysis should be the main goal of the review.

Minor comments

1. Lines 27-28. Keywords.

“Bolus Targeting Solution”

The authors should remove this term as they did not use or discuss it in the text.

2.

Line 85

“CGM”

And Fig. 2(b)

“CHO”

The author should explain these abbreviations.

3.

Lines 142-143

“The selection process is outlined in the PRISMA flow diagram (Error! Reference source not found.).”

Please correct this problem.

4.

Lines 219-220

“(a) Representation of models included in physioloigical models.”

In my opinion, this sounds somewhat casuistic. Please rephrase.

By the way, please change the word “physioloigical” to “physiological”.

5.

Figure 3(a)

“Gramatical Evolution”

Typo. Please change the word “Gramatical” to “Grammatical”

Round 2

Reviewer 2 Report

Comments and Suggestions for Authors

The authors took my comments into account. Congratulations to the authors for their good work and insightful review.